# Genome-wide identification and functional analysis of *ARF* transcription factors in *Brassica juncea* var. *tumida*

Wenbo Li[1], Fabo Chen[1], Yinping Wang[1], Haoyue Zheng[1], Qinqin Yi[1], Yun Ren[2], Jian Gao[1]*

**1** School of Advanced Agriculture and Bioengineering, Yangtze Normal University, Fuling, Chongqing, China, **2** College of Landscape Architecture and Life Science, Chongqing University of Arts and Sciences, Yongchuan, Chongqing, China

* Gaojian_genomics@163.com

**Data Availability Statement:** All relevant data are within the manuscript and its Supporting Information files.

## Abstract

Auxin signalling is vital for plant growth and development, from embryogenesis to senescence. Recent studies have shown that auxin regulates biological processes by mediating gene expression through a family of functionally original DNA-binding auxin response factors, which exist in a large multi-gene family in plants. However, to date, no information has been available about characteristics of the ARF gene family in *Brassica juncea* var. *tumida*. In this study, 65 *B. juncea* genes that encode ARF proteins were identified in the *B. juncea* whole-genome, classified into three phylogenetical groups and found to be widely and randomly distributed in the A-and B-genome. Highly conserved proteins were also found within each ortholog based on gene structure and conserved motifs, as well as clustering level. Furthermore, promoter *cis*-element analysis of *BjARFs* demonstrated that these genes affect the levels of plant hormones, such as auxin, salicylic, gibberellin acid, MeJA, abscisic acid, and ethylene. Expression analysis showed that differentially expressed *BjARF* genes were detected during the seedling stage, tumor stem development and the flowering period of *B. juncea*. Interestingly, we found that *BjARF2b_A*, *BjARF3b_A*, *BjARF6b*_A, and *BjAR-F17a_B* were significantly expressed in tumor stem, and an exogenous auxin assay indicated that these genes were sensitive to auxin and IAA signaling. Moreover, eight of the nine *BjARF10/16/17* genes and all of the *BjARF6/8* genes were involved in post-transcriptional regulation, targeted by *Bj-miR160* and *Bj-miR167c*, respectively. This analysis provides deeper insight of diversification for *ARFs* and will facilitate further dissection of *ARF* gene function in *B. juncea*.

## Introduction

Auxin signaling is crucial for plant growth and development from embryogenesis to senescence. Many biological processes are associated with auxin signaling, including growth and development of root/stem, formation and differentiation of vascular tissue, apical dominance, and stress responsiveness [1, 2]. The effects of many active synthetic auxins on gene expression and regulation have been explored in crops, and the results have indicated that two types of

**Funding:** This work was supported by Natural Science Foundation of CSTB (cstc2019jcyj-msxmX0652), Science and Technology Plan Projects of Fuling District (FLKJ, 2018BBB3009).

**Competing interests:** The authors have declared that no competing interests exist.

transcription factor families are required to regulate the expression of Auxin response genes: the *ARF* (Auxin response factor) family and the Aux/IAA repressor family [3]. *ARFs* act as transcription factors to control the expression of auxin- response genes by binding to auxin response element (AuxRE, TGTCTC motif) or its variations, such as TGTCCC, TGTCAC, and TGTCGG, resulting in the repression or activation of these target genes in promoters of primary or early auxin-responsive genes. An ARF protein usually consists of three modular and portable domain: a DNA-binding domain, such as the B3-like DNA binding domain in the N-terminal (DBD); a middle region (MR), which functions as a repression or activation domain via a yeast-two-hybrid confirmatory assay; and a carboxyl-terminal dimerization domain (CTD), which can mediate homo- and hetero-dimerization of ARF or hetero-dimerization of ARF and Axu/IAA proteins [4, 5]. However, a minority of genes lack protein domains, due to each domain being functionally independent, such as ARF3, ARF13 and ARF17 in *Arabidopsis*.

To date, numerous studies have characterized the *ARF* gene family based on genome-wide identification in many plant species, such as *Arabidopsis* [6], rice [7], maize [8], wheat [9], rapeseed [10], apple [11], barely [12], *Vitis vinifera* [13], tomato [14], *Populus trichocarpa* [15], *Brassica rapa* [16], and *Brassica napus* [17]. In *Arabidopsis*, 22 *ARF* genes (except one pseudogene) have been identified, which were categorized into three clades based on their amino acid sequence. Expression pattern analysis suggested that these genes are transcribed in a wide variety of tissues and plant development stages. However, 13 genes distributed on chromosome 1, which appear to be restricted to embryogenesis/seed development [18]. Additionally, combined with classical genetic approaches, the function of partial ARF family members' has been confirmed. Of these TFs, *ARF1* was the first identified TFs using a yeast-one-hybrid screen with a core motif, and has been found to respond to dark-induced senescence in leaves to regulate flower development [19, 20]. In *ARF2* single mutant, leaf senescence, flowering, and floral abscission were delayed, indicating that its function is mediated by light [21]. *ARF3* and *ARF4* are involved in reproductive and vegetative tissue development [22]. *ARF5* is associated with vascular strands and embryo axis formation [23, 24]. *ARF7* participates in the response of impaired hypocotyls to blue light [25, 26]. *ARF8* regulates fertilization and fruit development [27, 28]. In addition, T-DNA insertion mutants have shown that many *ARF* family members have overlapping functions in *Arabidopsis* [3, 29, 30]. In rice, 25 *OsARF* genes were identified using the *AtARF* protein sequence as a query for a BLAST search on the *Oryza sativa* genome [7]. Furthermore, nine OsARF proteins were predicted to function as activators and 16 as repressors. The transcription of *OsARF1*, *OsARF5*, *OsARF14*, *OsARF21*, and *OsARF23* was affected with auxin treatment under light conditions, and *OsARF1*, *OsARF2*, *OsARF16*, *OsARF21*, and *OsARF23* can be positively regulated by dark conditions [7]. An antisense phenotype revealed that *OsARF1* is essential for vegetative and reproductive development associated with growth retardation, low vigor, and sterility phenotype [31]. *OsARF8* regulates hypocotyl elongation and controls auxin homeostasis [7]. Furthermore, studies have shown that targeting of *ARF10/16/17* by *miR160* [32, 33] and *ARF6/8* by *miRNA167* [34, 35] is indispensable for various aspects of development, which might highly conserved in plants.

Tumorous stem mustard (*Brassica juncea* var. *tumida* Tsen et Lee), is one of the most important economic crops, and the raw material for Fuling mustard, in China. How to improve the yield (Tumor stem, the vegetative organ) of this crop is a key issue for the Chinese pickles industry. Auxin is vital for plant growth and development, exploring whether or not the effect of the auxin signaling pathway in crop production is meaningful. *ARF* gene family exists in tumorous stem mustard, but no evidence is available on the genome-wide identification and functional analysis of *ARF* genes in tumorous stem mustard, due to a lack

of genome information. Genetic divergence between *Arabidopsis* and rice investigated by genome-wide analysis revealed that most *OsARFs* are related to *AtARFs*, and form sister pairs [3, 7]. The first assembly of *B. juncea* genome data has recently been published [36], making it possible and feasible to isolate functional gene families from *B. juncea* genome. Despite their crucial role in plant development, *ARF* family genes in *B. juncea* have not yet been identified and analyzed in detail. In this study, we identified *ARF* genes in *B. juncea* genome, determined their chromosomal distribution, gene and protein structure, and confirmed the expression of 65 *B. juncea ARF* genes by qRT-PCR. Furthermore, we analyzed *cis*-acting elements for *BjARF*-gene promoter regions and predicted potential targets for small RNA. Moreover, the phylogenetic relationship of *ARFs* between *Arabidopsis*, *B. napus*, and *B. juncea* were also compared, and partial specific *BjARF* gene expression was validated with an auxin treatment assay. We hope that this work will be helpful for more functional investigations of *ARFs* in tumorous stem mustard in future.

## Methods

### B. juncea *ARF* gene identification and chromosomal distribution

The *B. juncea* genome database was kindly supported by Zhejiang University. A local implementation of PlantTFDB V4.0 BLAST was used for sequence searching [37]. All known *ARFs* from *A. thaliana* (22 *ARF* genes, except a pseudogene) and *B. napus* (64 *ARF* genes) were used in initial protein queries. Gene names and IDs are listed in S1 Table, and *BjARF* genes were named/classified based on *A. thaliana ARF* genes. The hidden Markov model (HMM) profile (PF06507) of *ARF* family genes can be downloaded from PFAM (http://pfam.xfam.org/) [38], and the hmm build command from HMMER 3.0 software package can be used to transform it into a hidden Markov model sequential pattern. Then, we queried the "Yonganxiaoye" *B. juncea* genome database, with the following search parameters were followings: BLASTP, E value < 1e-10, identity > 70%, query coverage > 95%, and other parameters were defaulted. All non-redundant hits with expected values were collected and then compared with the ARF family in PlnTFDB [39] and PlantTFDB [37]. Subsequently, each candidate *ARF* gene was further confirmed base on the presence of the Auxin_resp domain using SMART [40], CDD [41] and Inter-ProSca. The molecular weights and theoretical isoelectric points of the *BjARFs* were determined within Expasy [42]. Information on the physical location of the *ARFs* in *B. juncea* were collected from the corresponding GFF files, and mg2c was used to visualize the distribution of *ARF* genes on each *B. juncea* chromosome.

### Phylogenetic analysis of *BjARFs*

Phylogenetic analysis of multiple sequences was performed using full-length protein sequences of *ARFs* with ClustalW alignment. Then a phylogenetic tree was constructed with MEGA 7.0 software [43] by the Neighbor-Joining (NJ) method, carried out with 1,000 replicates bootstrap test.

### Gene structure and conserved motif of *BjARFs*

The exon/intron structures of the *BjARFs* were illustrated from alignments of genomic sequences and cDNA, and drawn with GSDS 2.0 [44]. Inputting the GFF files, MEME (suite 5.0.2) [45] was used to elucidate conserved motifs of *BjARFs*, and ran locally with the following parameters: maximum number of motifs -9, and width of motif constrained from 10 to 300. MAFFT Version 7 [46] was selected to present the Auxin_resp domain.

### *cis*-acting elements analysis in promoter regions

To investigate the putative *cis*-acting elements of 65 *BjARFs* candidate genes, *BjARF* promoter sequences (2000 bp upstream of the initiation codon 'ATG') were extracted from genome database of the *B. juncea*. PlantCARE [47] was used to obtain *cis*-acting elements.

### Gene expression profile analysis

To analyze the expression profiles of *BjARFs*, we used a privately available RNA expression profile data, including leaf and stem of seedling stage (SS, 20-days after germination), mature stage (MS, 90-days after germination) and flower stage (FS, 150-days after germination) from three different cultivars, which named yonganxiaoye1 (YA1), yonganxiaoye2 (YA2) and yonganxiaoye3 (YA3). We used Log2 (FPKM) to calculate levels of *BjARF* expression, and their expression patterns were clustered by hierarchical clustering models and illustrated using a homemade R programming language.

### Prediction of *BjARFs* targeted by *miRNAs*

*BjARFs* targeted by *miRNA* collected from the literature were predicted using psRNATarget [48], and the parameters were set as follows: maximum expectation of 3 and target accessibility (UPE) of 50.

### Auxin treatment and tissue preparations

"Yonganxiaoye1" seeds were planted in field, and 10-days-old seedlings were transplanted into flower plots with nutritional soil substrate, and grown under natural conditions (16~24 ˚C with 16 h light/8 h dark). Root, leaf, tumor stem, and flowering bud tissues were collected at different growth stage. For the IAA treatment, the plant root was irrigated with 100 mL solution and plant surface was sprayed using an atomizer, two sets of seedling (20-days-old) and pre-stigmas (60-days-old) plants were irrigated/sprayed by water with and without 50 μM IAA, respectively. And then keep it for eight hours, tumor stem and leaf tissues were collected. All plant tissues were frozen in liquid nitrogen and subsequently stored at -80 ˚C until RNA isolation.

### Isolation of total RNA, reverse-transcription and real-time RT-PCR analysis

RNA was isolated using the TRIzol method (Invitrogen, USA) and reverse transcribed to cDNA using M-MLV transcriptase (Roche, Switzerland). qRT-PCR was carried out using SsoAdvancedTM SYBR Green supermix (Bio-Rad, USA) on a Bio-Rad CFX96TM Real Time PCR System according to the manufacturer's instructions. The relative quantitative method ($2^{-\Delta\Delta CT}$) [49] was used to calculate the fold change in the expression of target genes, and the *B. juncea β-Actin* gene was used as the internal control (S4 Table). Each measurement was made using two biological samples in triplicate per sample.

### Statistical analysis

Statistical analysis was performed using paired-samples *t*-test (one-tail) by SPSS 19.0 software, the significant differences of gene relative expression levels was detected between the treatment and control.

## Results

### Genome-wide identification and chromosomal localization of *ARF* genes in *B. juncea*

To identify genes encoding *ARF* transcription factors in *B. juncea*, the Auxin-resp domain (A conserved region of auxin-responsive transcription factors, PF06507) was used to search against *B. juncea* proteins in HMMER software. All non-redundant hits with expected values were collected and then compared with the *ARF* family in PlnTFDB and PlantTFDB website. In addition, we verified the candidate *ARF* genes through the presence of the Auxin-resp domain using SMART, Prosca, and CDD. In total, 65 *ARF* genes were identified in *B. juncea* (S1 Table). Of those, we accurately named *BjARF* genes following their closest orthologs in *A. thaliana* and coded their different paralogs as a, b, c, etc., together with the order of the homologous chromosomes (S1 Table). The result shows that length of 65 ARF proteins varies from 157 to 1197 amino acids with predicted molecular weight of 17.85 to 132.65 kDa in *B. juncea* (S2 Table). In addition, analysis of chromosome location showed that those *BjARF* genes were distributed on all chromosomes present in the *B. juncea* genome (A1-A10, B01-B08), as well as seven contigs and three scaffolds comprised of Contig71, Contig94, Contig157, Contig533, Contig705, Contig1207, Contig2271, Super_scaffold_57, Super_scaffold_127 and Super_scaffold_187. We found that the numbers of *BjARF*s mapped on each chromosome were uneven and ranged from one to seven (Fig 1, S2 Table). In other species, segmental duplication, tandem duplication, and polyploidization were identified determined the genomic locations of the ARF gene family. We identified 26 pairs of segmentally duplicated *ARF* genes in the *B. juncea* genome (S1 Fig).

### Phylogenetic relationships of *ARF* genes in *B. juncea*, *A. thaliana*, and *B. napus*

In total, 150 *ARFs* (including 65 obtained from this study in *B. juncea*, 22 from *A. thaliana* and 63 from *B. napa*) were used to construct a NJ phylogenetic tree by MEGA 7.0 software. As shown in Fig 2, we found that those *ARF* genes were clustered into three groups, named from I to III. In addition, the *ARF* orthologs were clearly distinguished in *B. juncea*. *BjARF10*, *BjARF17* and *BjARF16* were categorized into group I. *BjARF4*, *BjARF3*, *BjARF6*, *BjARF8*, *BjARF5*, *BjARF7* and *BjARF19* were categorized into groupII. *BjARF1*, *BjARF2*, *BjARF18*, *BjARF11* and *BjARF9* were clustered in group III. However, *ARF12*, *ARF13*, *ARF14*, *ARF15*, *ARF20*, *ARF21 and ARF22* were not identified in *B. juncea* in this study, *AtARF12*, *AtARF13*, *AtARF14*, *AtARF15*, *AtARF20*, *AtARF21*, and *AtARF22* were categorized into a sub-group of Group III, and partial *BjARFs* and *BnARFs* were categorized into a novel sub-groups of Group III (Fig 2, S2 Fig), named *ARFun1*, *ARFun2* and *ARFun3*, respectively.

### Gene and protein structure of *ARFs* in *B. juncea*

To demonstrate structural diversity of *BjARFs*, the conserved motif and exon/intron structure from *BjARFs* promoters were analyzed. An unrooted NJ tree was constructed using 65 ARF protein sequences from *B. juncea* (Fig 3A). Their gene structures were analyzed by GSDS 2.0 and are displayed in Fig 3B. The results showed that Auxin_resp domain was located in the N-terminal of all ARF genes, and the number of introns varies from one (*BjARF17*) to 17 (*BjARF18b_O*). Most genes within the same sub-family shared a similar structure and length, those in group I contained one to five introns less than those in group II and III, and members of group III had longer introns than members of group I and II, especially for *BjARF11a_B*, *BjARF18b_O*, *BjARFun1a2_B*, and *BjARFun2a_A*. Structural variation in genes provides

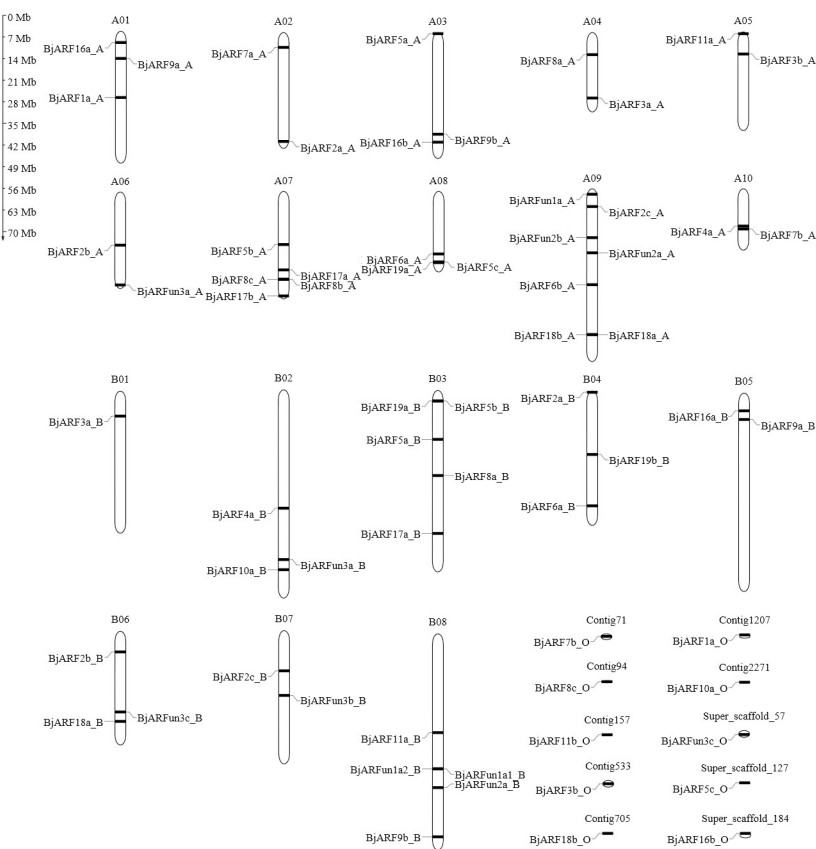

**Fig 1. Distribution of *ARF* genes in the *Brassica juncea* genome.** Chromosomal distribution of *ARF* genes in *B. juncea* was determined, and the locations of closely linked genes are shown. The chromosome number is indicated on the top of each chromosome. Scale = megabases (Mb).

valuable information on duplication events within gene families, these results suggest that different subfamily genes represent different functions in *B.juncea* development.

Furthermore, to better understand the sequence characteristics of *BjARF* genes, MEME was used to predict motifs based on 65 BjARF protein sequences, nine motifs named 1–9 were identified (Fig 4, S3 Fig). These conserved motifs contained between 21 (motif 8) and 60 (motif 1) amino acids. Each BjARF protein contain two (BjARF10a_B) to nine conserved motifs, with most containing nine motifs. Interestingly, most of the BjARF proteins contained motif 1 in the Auxin_resp domain. Moreover, we presented the Auxin_resp domain structures by constructing multiple alignments of all 65 BjARF proteins using MAFFT version 7 (S4 Fig). We concluded that similar gene structures and the motif architecture of *BjARF* orthologs were significantly clustered into the same phylogenetic tree, it suggesting they may have different roles in *B. juncea* development.

## *Cis*-acting elements prediction of *BjARF* gene promoter regions

*Cis*-acting elements are vital sequences that are targeted by transcription factors to regulate gene expression. To investigate the putative functions of 65 *BjARF* genes in *B. juncea*, we obtained 2000 bp sequences upstream of the initiation codon 'ATG' and then used PlantCARE to predict *cis*-acting element. More than 234 types of putative *cis*-acting elements were identified in the promoter of *BjARFs* (S3 Table), including light responsive elements (G-box, TCT-

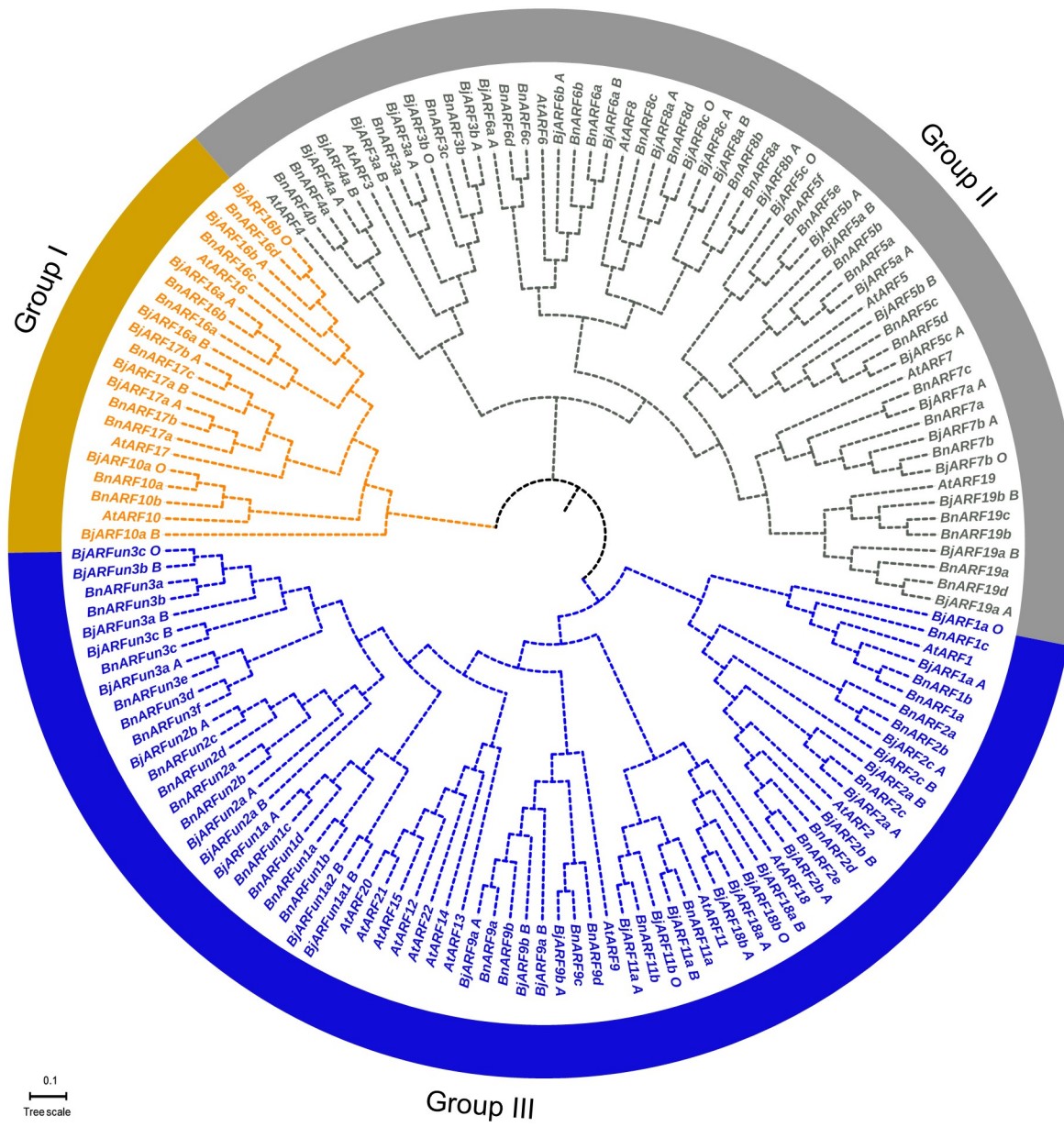

**Fig 2. Phylogenetic relationships of *ARF* genes from three different species (*B. juncea*, *Arabidopsis thaliana*, and *Brassica napa*).** The phylogenetic tree was constructed using MEGA 7.0 via the Neighbor-Joining method with 1,000 bootstraps. Different groups of the *ARF* family are represented in different colors.

motif, GT1-motif, AE-box, Box 4, GATA-motif, I-box, TCCC-motif, MRE, GA-motif, 3-AF1 binding site, chs-CMA1a, LAMP-element, Box II, ATCT-motif, ATC-motif, GTGGC-motif, ACE, chs-CMA2a, Gap-box, GATT-motif, AT1-motif, CAG-motif, Sp1, chs-CMA2b, chs-Unit 1 m1, and L-box), promoter/enhancer elements (CAAT-box, TATA-box, CCAAT-box, A-box, AT-rich element, Box III, Unnamed__1, AT-rich sequence, 3-AF3 binding site, Box II -like sequence, MBSI, and HD-Zip 3), phytohormone response element [such as abscisic acid (ABRE), gibberellin (GARE-motif, P-box, TATC-box), MeJA (TGACG-motif), salicylic acid and auxin (TCA-element) and auxin (TGA-element, AuxRR-core, AuxRE)] (Fig 5), stress

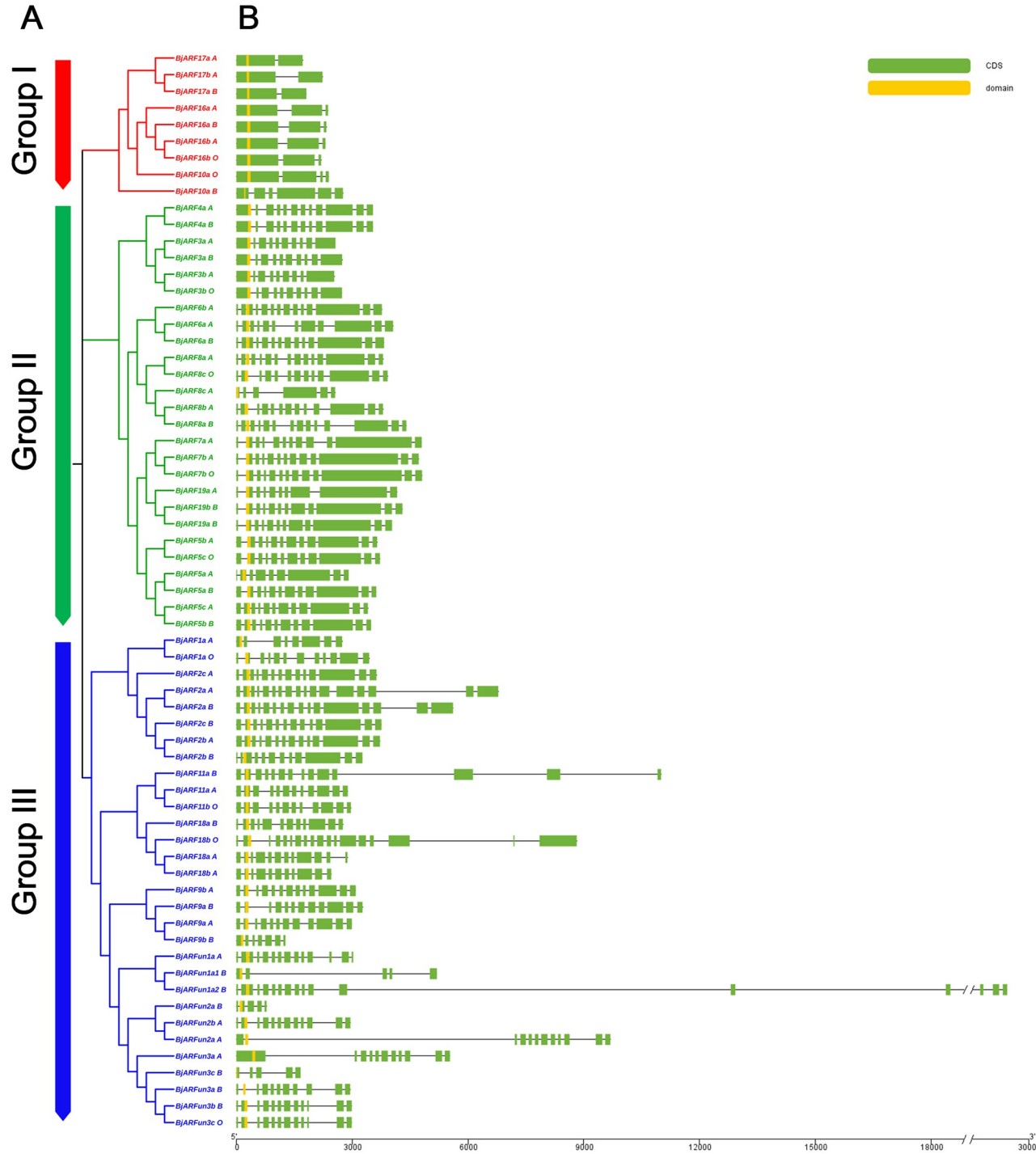

**Fig 3. Phylogenetic tree of the *ARF* genes and genomic organization of *ARF* genes in *B. juncea*.** A. NJ tree of *BjARFs*; the Auxin_resp domain was identified and the phylogenetic tree was constructed using MEGA 7, via the Neighbor-Joining method with 1000 bootstraps. Different groups of the *ARF* family are shown in different colors. B. Exon and intron structures of *BjARFs*, CDSs are shown in green, black lines connecting two CDSs represent introns.

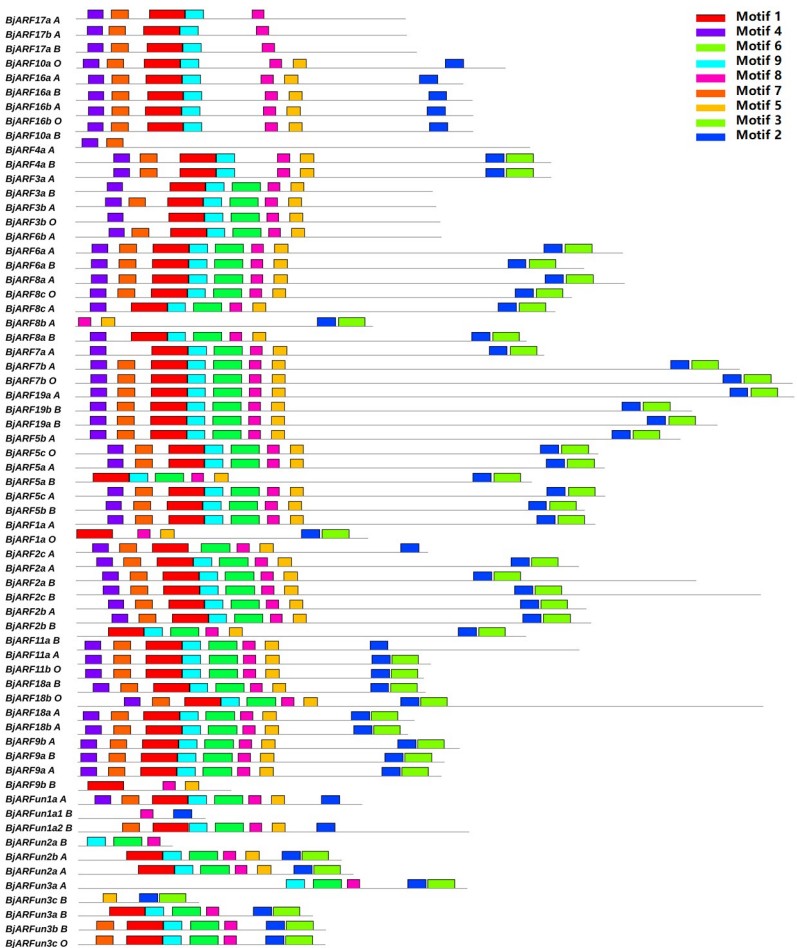

**Fig 4. Conserved motifs of *BjARF* proteins identified in this study.** The conserved motifs were identified using MEME (suite 4.11.4) based on *BjARF* protein sequences, and each motif is indicated with a colored box numbered 1 to 9 at the upper right corner.

response elements (ARE, MBS, TC-rich repeats, LTR, WUN-motif and GC-motif), development and tissue specificity elements (CAT-box, O2-site, GCN4_motif, HD-Zip 1, RY-element, AACA_motif and a unnamed motif) and circadian control elements (circadian and MSA-like motif) (S3 Table). This suggests the most *BjARF* genes are involved in different biological processes, including responses to phytohormones, abiotic stresses and tissue development in *B. juncea*.

## Prediction of small RNA for potential targets

To delineate the *miRNA*-mediated posttranscriptional regulation of *BjARF* genes, we searched the coding regions and 3′ UTRs of all *BjARFs* for the targets of *Bj-miR160* and *Bj-miR167*. The results showed that eight *BjARF* genes belonged to Group I that are complementary to the *BjmiR160* mature sequences (Fig 6A), In addition, eight *BjARF* genes belonged to Group II that are complementary to the *Bj-miR167c* mature sequences (Fig 6B). The results suggest that *BjmiR160* and *BjmiR167c* may specifically target these *BjARF* genes in *B. juncea*. *BjmiR160* was located in the coding region of *BjARF16a_A*, *BjARF16a_B*, *BjARF16b_A*, *BjARF16b_O*,

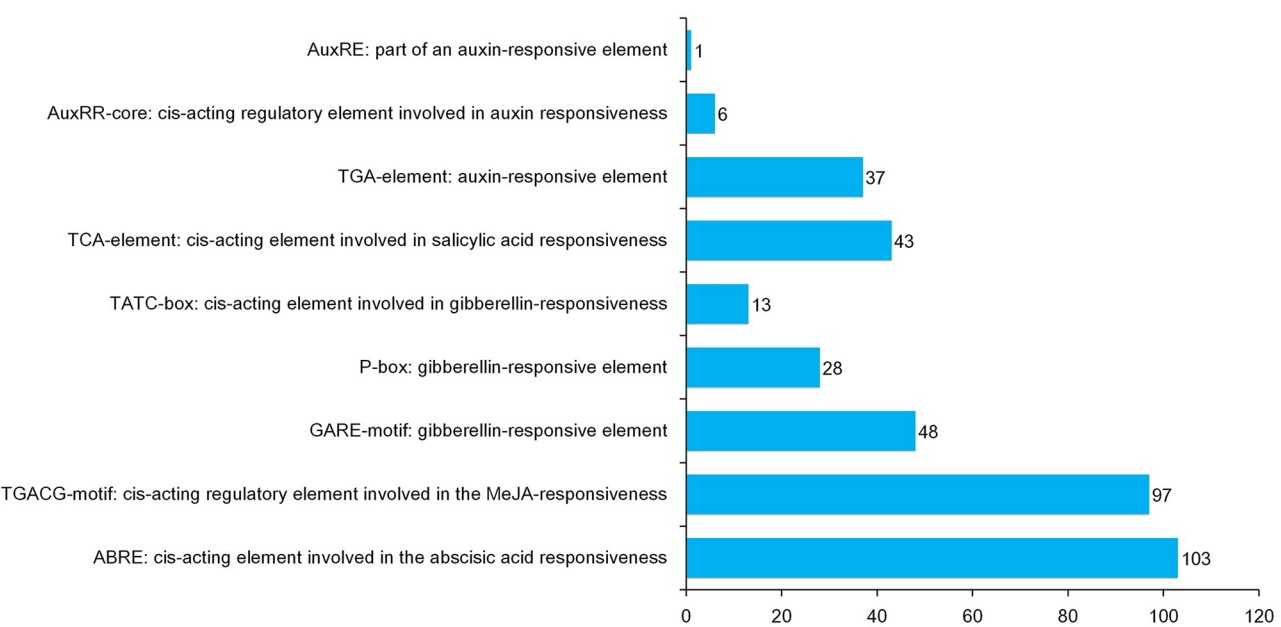

**Fig 5. Analyze of *cis*-acting elements response to phytohormones in *BjARFs* promoters.** Value of x-axis means the number of *cis*-acting elements which founded in *BjARFs* promoters.

*BjARF17a_A*, *BjARF17a_B*, *BjARF17b_A*, and *BjARF10a_O* (Fig 6A), and *BjmiR167c* was located in the coding region of *BjARF6a_A*, *BjARF6a_B*, *BjARF6b_A*, *BjARF8a_B*, *BjARF8b_A*, *BjARF8c_A*, *BjARF8c_O*, and *BjARF8a_A* (Fig 6B), respectively. These results revealed that post-transcriptional control of *ARFs* mediated by *Bj-miR160* and *Bj-miR167c* are conserved in plants.

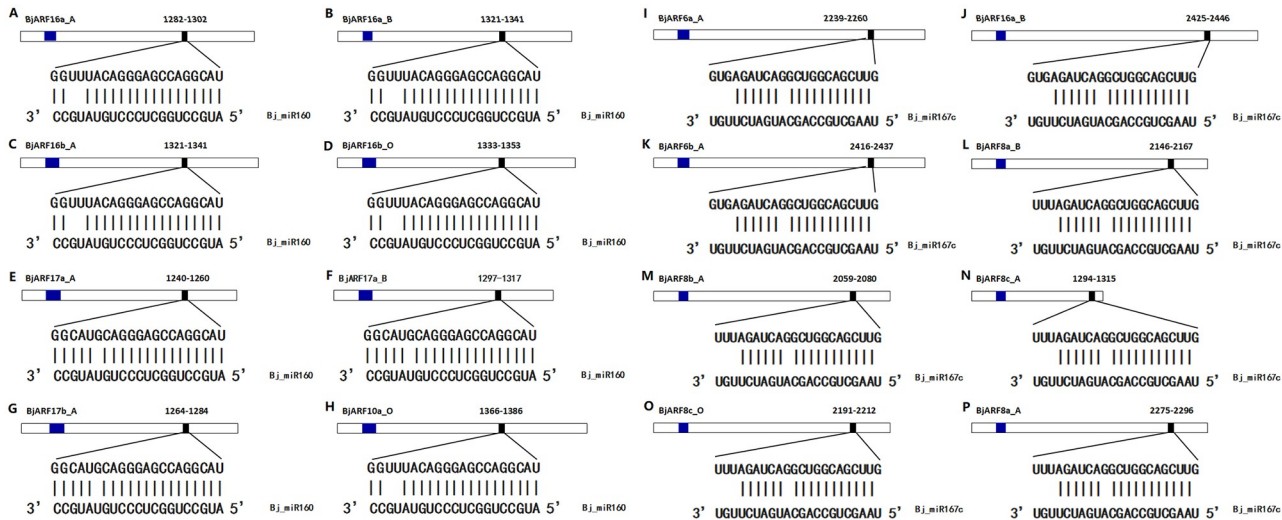

**Fig 6. *miR160/167c*-mediated post-transcriptional regulation of *BjARFs*.** (A-H). Prediction of the *BjARFs* regulated by *Bj-miR160*; (I-P). Prediction of the *BjARFs* regulated by *Bj-miR167c*. *miRNA* target sites (Black) with the nucleotide positions of *BjARF* transcripts are shown, and the Auxin_resp domain (Blue) with the nucleotide positions of *BjARF* transcripts are shown. The RNA sequences of each complementary site from 5′ to 3′ and the predicted *miRNA* sequence from 3′ to 5′ are indicated in the expanded regions.

## Expression profiles of *BjARF* genes in *B. juncea*

A certain levels of gene expression is vital for determining gene function. Therefore, we analyzed the expression profiles of all identified 65 *BjARFs* available from our private RNA expression profile data. A heat map showing the expression of *BjARFs* was constructed using homemade R and is shown in Fig 7A. *BjARF* genes were divided into four groups based on their expression profiles: *BjARF10a_O*, *BjARF1a_A*, and *BjARF7a_A* were highly expressed in all tissues during the leaf of seedling stage (SS), leaf and stem of flowing stage (FS), leaf and stem of mature stage (MS) from YA1 (yonganxiaoye1), YA2 (yonganxiaoye2), and YA3 (yonganxiaoye3) respectively. Interestingly, *BjARF10a_O* was highly accumulated in the stem of flowing period (FS) and mature stage (MS), and *BjARF1a_A* was preferentially expressed in leaf of FS and MS. In addition, *BjARF2b_A* and *BjARF6b_A* were found specifically expressed in stem from YA2 and YA1 of FS and MS, respectively. Further, *BjARF2b_A*, *BjARF2a_B* and *BjARF16b_O* were specifically expressed in leaves at different periods of *B. juncea* development. However, no other *BjARF* genes were significantly detected in all tested tissues (Fig 7A).

These candidate *BjARF* genes, which may be significantly expressed in different tissues, were validated using qRT-PCR. Plant seedlings (20-days after germination), plant leaves in the vegetal period, tumor stem of vegetal period (60-days after germination, VP), and flower period (150-days after germination, FP), flower and legume of Yonganxiaoye 1 tissue were collected for RNA isolation. Analysis of expression patterns showed that, *BjARF16b_O* and *BjARF2b_A* were significantly expressed during early plant growth in leaves (Fig 7E and 7F),

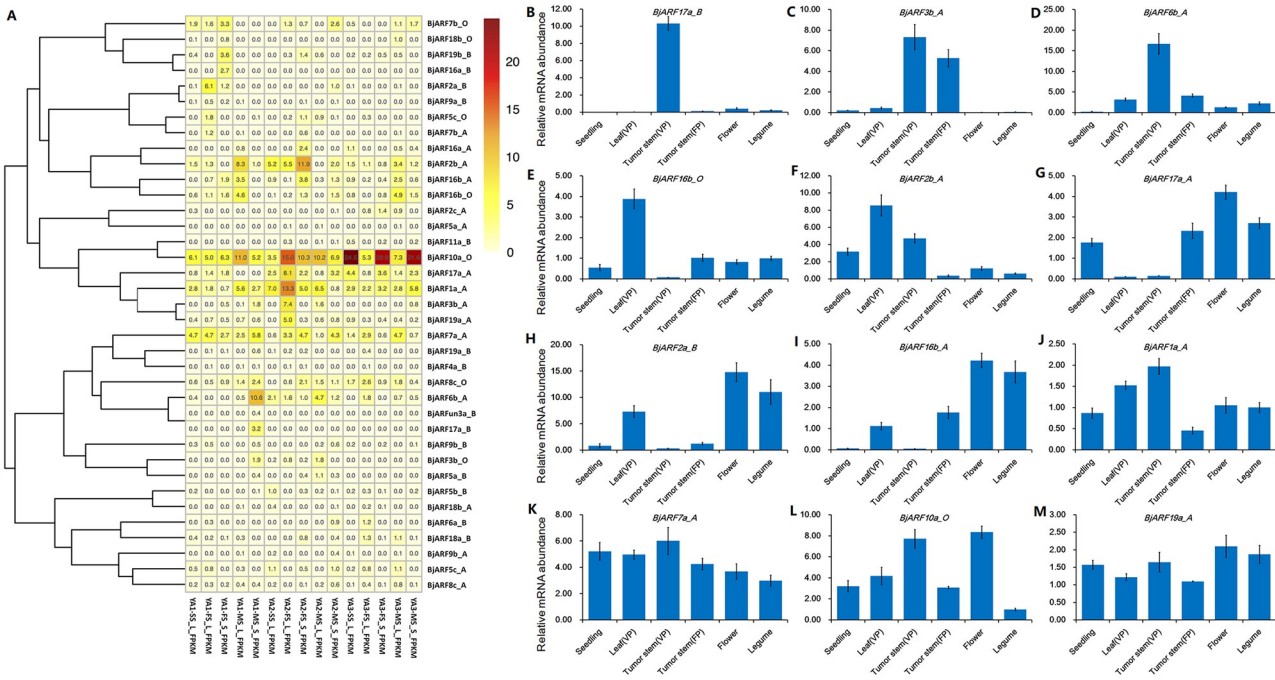

**Fig 7. Expression pattern of *BjARFs*.** A. Expression profiles of *BjARFs* from RNA-seq in different tissues and cultivars. Color represents *BjARF* expression levels: Log2 (FPKM). The phylogenetic relationship is shown on the left. RNA expression data were selected as follows: leaf (L) of the seedling stage (SS), leaf and stem (S) of the flowing period (FP), leaf and stem of the mature period (MP) from YA1 (yonganxiaoye1), YA2 (yonganxiaoye2) and YA3 (yonganxiaoye3), respectively. B–M. Validation of candidate *BjARFs* using qRT-PCR, based on altered expression levels. From left to right representing seedling, leaf (VP), tumor stem (VP), tumor stem (FP), flower, and legume. Error bars show the standard error calculated from three biological replicates. Values are presented as the mean ± SEM.

*BjARF3b_A* and *BjARF6b_A* were highly expressed in the tumor stem (Fig 7C and 7D), and *BjARF17a_A*, *BjARF2a_B*, and *BjARF16b_A* were significantly expressed during the reproductive stage of *B. juncea* (Fig 7G–7I). Only *BjARF17a_B* was exclusively expressed in the stem nodule during *B. juncea* stem tumor development (Fig 7B). However, expression of *BjARF1a_A*, *BjARF7a_A*, *BjARF10a_O*, and *BjARF19a_A* was widespread (Fig 7J–7M).

## Induction of *BjARF* gene expression by exogenous auxin

To confirm whether candidate genes are induced by auxin signal, 20-days-old seedlings and 60-days-old plants were treated with and without 50 μM IAA for 8 h. The seedlings, plant leaf, and tumor stem were collected and relative gene expression of eight candidate *BjARF* genes significantly expressed in different plant tissues was determined. The results showed that the expression of most *BjARF* genes was induced in specific organization and development period of plant. For example, expression of *BjARF2b_A*, *BjARF2a_B*, *BjARF16b_A*, and *BjARF16b_O* were significantly increased in leaf exposed to 50 μM IAA (Fig 8A, 8E, 8F and 8H); expression of *BjARF3b_A*, *BjARF6b_A* and *BjARF17a_B* were significantly increased in tumor stem under exogenous auxin treatment (Fig 8B–8D). In contrast, expression of *BjARF17a_A* was induced following with treatment with exogenous auxin in seedlings (Fig 8G). These results suggest that *BjARF* genes are sensitive to auxin, and that *BjARFs* play crucial roles during *B. juncea* development and are regulated by endogenous plant hormones.

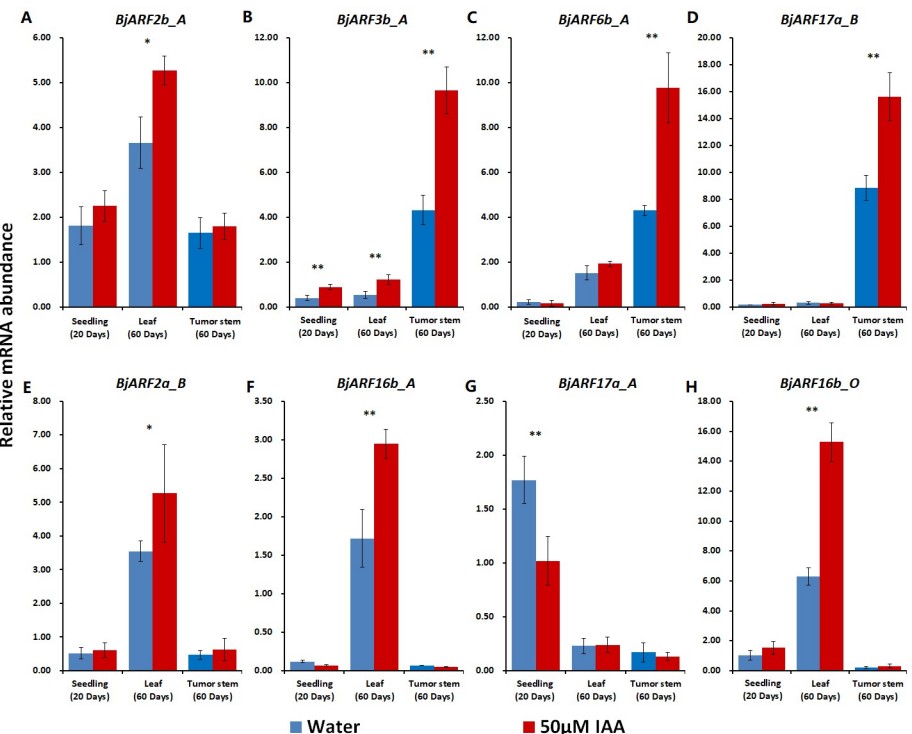

**Fig 8. Fold induction of *BjARF* genes in response to exogenous auxin.** For *B. juncea* Yonganxiaoye 1, 20-day-old seedling and a 60-day-old leaf and tumor stem were collected following treatment with 50 μM IAA and water as a control. Values represent the mean ± standard error of the mean (SEM). Paired-samples *t*-test (one-tail) for 50 μM IAA treatment with a negative control to detect significant differences, $^{**}$ p < 0.01 and $^{*}$ p < 0.05.

## Discussion

The theory of U's triangle illustrates the evolutionary relationship of the cruciferous plants *B. nigra*, *B. oleracea*, and *B. rapa*. U's triangle showed that *B. juncea* originated from *B. rapa* and *B. nigra* through hybridization and genome duplication [50]. *B. juncea* var. *tumdia* is a popular vegetable in China, and becoming geographical indication products of Chongqing city. In recent years, *ARF* genes have gained widespread attention due to the large number of family members and their response to most of life processes in plants, and *ARFs* were genome-wide identified in Arabidopsis, rice, maize and wheat successively [6, 7, 8, 9]. In this study, we characterized 65 *B.juncea ARF* genes by genome-wide analysis (S1 Table), which is a higher number than reported in other species. We suspect that this may be due to plant evolution and by *B. juncea* belonging to an allotetraploid. In recent years, researches have demonstrated that segmental duplications were existed and play vital roles in the gene expansion of *ARF* transcription factor gene family in many plants [16, 17]. Here, 26 pairs of segmentally duplicated *ARF* genes were identified in the *B. juncea* genome. Phylogenetic analysis revealed that each *ARF* genes was conserved in Cruciferae [6, 16, 17], especially between *B, juncea* and *B. napa*. We named the *BjARFs* according to the homologous *ARF* gene in *Arabidopsis* (S1 Table). Interestingly, we found that *AtARF12–15* and *AtARF20–22* were categorized into a sub-group of Group III, and 11 *BjARFs* and 14 *BnARFs* were categorized into another sub-group of Group III. Thus, we speculate that *B. juncea* evolved a set of unique *ARF* genes to adapt its development compared to Arabidopsis, and that *B. napa* was more closely related to *B. juncea* in evolution. It suggested that gene function of unique genes are more similar between *B. juncea* and *B. napa*, we should give priority to *B. napa* as a reference for gene function research in future. These unique genes may present functional specificity in different plant species, explaining the existence of a larger family in *B. juncea*. In addition, gene structures and conserved protein motifs in *BjARFs* were found in different *ARF* orthologs with a diverse number of introns, we speculated that different *ARF* orthologs shared similar functions.

Analysis of *BjARF* gene expression showed that these genes could be divided into four groups, and partial *BjARF* members were significantly expressed in plant tissues or different varieties. Many *ARFs* (*AtARF2*, *AtARF7*, *AtARF19*, *OsARF1*, *OsARF5*, *OsARF19*, *OsARF24*, *SlARF2*, and *SlARF19*) were found to be involved in regulating the development of plant roots, stems, leaves and flowers [9, 20, 31, 51, 52]. In this study, 12 candidate genes were subjected to expression pattern analysis. Interestingly, we obtained two *BjARF* genes that were significantly expressed in plant seedlings and leaves during early plant development, three *BjARF* genes significantly expressed in plant flowers and legumes that may be associated with plant reproductive and embryonic development, four *BjARF* genes that were specifically expressed in stem during stem development. In addition, four *BjARF* genes were widely expressed in the whole plant during development process. Therefore, it is speculated that *BjARF* members with the same expression patterns share similar biological functions. For example, *BjARF17a_B*, *BjARF3b_A*, *BjARF6b_A*, and *BjARF16b_O* may play key roles in stem tumor intumescence.

In plants, *miRNAs* play a regulatory role in plant cell by negatively affecting gene expression at the post-transcriptional level. A number of pairs of microRNAs and their respective mRNA targets were predicted by database analyses [53]. Among these, *miR160* was shown to target *ARF10/16/17* and *miR160* was shown to target *ARF6/8*. We proposed that *miRNA160/167* might also response to auxin signaling via the negative regulation of auxin response factors in *B. juncea*, and we demonstrated that 16 *BjARF* members were putative targeted by *miR160/167*. In *Arabidopsis*, transgenic expression of *miR160*-resistant of *ARF10*, *ARF16* and *ARF17* resulted in pleiotropic phenotypes associated with their over-accumulation [54, 55]. The

results suggest that *miR160* is essential for seed germination, root-cap formation and in vitro shoot regeneration. In soybean, *miR160* was confirmed as a link between auxin and cytokinin action, by balancing auxin activity, through regulation of target repressor ARFs, for proper nodule development [56]. In tomato, *miR160* was found to be regulate auxin-mediated ovary patterning as well as floral organ abscission and lateral organ lamina outgrowth [57]. Previous studies have showed that *miR167* target auxin response element (AuxRE, TGTCTC motif or other similar sequences) and response to auxin signaling in early auxin-inducible gene promoters [58], and current study confirmed that *NtMIR167a-NtARF6/8* is critical for plants in regulating Pi starvation tolerance [59]. We believe that the mechanism of *miRNA160/167* involved in Auxin-mediated regulates mustard development will be revealled in our near further study.

Furthermore, we confirmed that eight significantly expressed candidate genes respond to exogenous hormones. When plants were exposed to IAA, expression of most of these *BjARF* genes was induced during a specific organization or development period of the plant (Fig 8). The relative expression of *AtARF4*, *AtARF16*, *AtARF19*, *OsARF1*, and *OsARF23* increased in response to auxin [8, 21–24, 31], while that of *ARF5*, *ARF14*, and *ARF15* decreased [23, 24]. In this study, most of the *BjARFs* were induced by exogenous auxin and auxin responsive elements were detected in promoter regions (S3 Table). In addition, the expression of *BjARFs* may also be regulated by miRNA, *AtARF6* and *AtARF8* were reported to be targets of *miR167* during the regulation anther and ovule development [53, 60–62], while *AtARF10* and *AtARF16* were reported to be targets of *miR160* during root cap formation [29, 53]. In present study, *BjARF10*, *BjARF16*, and *BjARF17* were predicted to be target genes of *Bj-miR160* (Fig 6A), while *BjARF6* and *BjARF8* were predicted to be target genes of *Bj-miR167c* (Fig 6B), suggesting that *Bj-miR160* and *Bj-miR167c* may participate in regulation of *B. juncea* growth and development by responding to changes in ARF gene expression. Therefore, the mechanism underlying the auxin-mediated regulation of *BjARFs* needs to be clarified, and regulation of the expression of other *BjARF* genes should be further explored.

Accumulating evidence suggests that auxin signaling is mediated by *ARFs*, is usually involved in plant morphogenesis, such as apical dominance, tropic responses meristem elongation, root development, and shoot elongation, as well as responses to various abiotic stresses [1, 2]. *AtARF1* and *AtARF2* were confirmed to control leaf senescence, silique ripening, and floral organ abscission independently of the ethylene and cytokinin response pathways in *Arabidopsis* [20]. In this study, *BjARF2b_A* and *BjARF16b_O* was significantly expressed during early plant growth in leaf; *BjARF2b_A*, *BjARF2a_B*, *BjARF16b_A*, and *BjARF16b_O* expression were conducted by exogenous auxin in leaves. Additionally, a recent study indicated that *AtARF3* integrates the functions of *AG* and *AP2* in floral meristem determinacy, while *AtARF4* is associated with organ polarity. *ARF5* is required for embryonic roots and flowers [22–24] and *ARF8* regulates fertilization and fruit development. ARF2, ARF4, ARF5, and ARF17 was confirmed be associated with anther development, pollen formation and gametophyte development [63, 64]. *ARF18* affects the yield of rapeseed through regulates seed weight and silique length [10]. At present study, *BjARF17a_A*, *BjARF2a_B*, and *BjARF16b_A* may response to regulate inflorescence development and reproductive in tumorous stem mustard, which is expressed extremely high during reproductive stage of *B. juncea* (Fig 7G–7I). Furthermore, we have obtained three candidate genes may associate with tumor stem development, it may provide new insight for yield improvement of tumorous stem mustard (Fig 7B–7D). In summary, 65 *BjARFs* were identified in *B. juncea*, and together with phylogenetic analysis, GSDS analysis, expression pattern analysis, and exogenous IAA treatment, the functions of partial *BjARF* gene members were predicted. Notably, some genes may regulate tumor intumescence, which is crucial for *B. juncea* production. The potential functions of other family members need to be evaluated and our speculations should be further explored. This would be helpful for future

research to focus on the functional analysis of tissue and development-dependent *ARF* transcription factors.

## Conclusions

In this study, a total of 65 *ARF* genes were identified and distributed on all chromosomes present in the *B. juncea* genome. All the family members have similar gene structures and protein domains to the auxin response factors of *Arabidopsis*. Several phytohormone response elements were found in the promoter region of *BjARFs*, indicating these genes may regulate plant development, which could benefit from a more targeted follow-up study. *Bj-miR160* and *Bj-miR167c* were predicted to regulate *BjARFs* gene expression through post-transcriptional, which is conserved in plants. Gene expression analysis showed that some *BjARF* family genes exhibit a special expression pattern and inducted by exogenous auxin, and we speculated that some family members may be involved in the regulation of tumorous stem mustard yield. Together, this work will be helpful for more functional investigations of *ARFs* in tumorous stem mustard in future.

## Supporting information

**S1 Table. Gene name and ID of *ARFs* in *B. juncea*, *Arabidopsis* and *B. napus*.** (XLSX)

**S2 Table. Characterization of *ARF* family genes in *B. juncea*.** (XLSX)

**S3 Table. *Cis*-acting elements of *BjARF* genes in promoter regions.** (XLSX)

**S4 Table. List of primers for qRT-PCR analysis of candidate *BjARFs*.** (XLSX)

**S5 Table. Expression patterns of global expression gene in various tissues of *Brassica juncea* var. *tumida*.** Including leaf of Seedling stage (SS), leaf and stem of flowering stage (FS), leaf and stem of mature stage (MS) from YA1 (yonganxiaoye1), YA2 (yonganxiaoye2) and YA3 (yonganxiaoye3) respectively. (XLSX)

**S1 Fig. Phylogenetic tree of the *ARF* gene family in *B. juncea* annotated with collinear and tandem relationships.** Curves connecting pairs of gene names suggest the collinear relationship. This annotated tree is output from 'family tree plotter' of MCscanX software. (PNG)

**S2 Fig. Phylogenetic relationship of spcial *ARF* genes from *B. juncea*, *A. thaliana*, and *B. napa*.** The phylogenetic tree was constructed using MEGA 7.0, the maximum likelihood method with 1,000 bootstraps. The different special of the *ARF* family are represented in different colors. (PNG)

**S3 Fig. Motif Logos of 9 conserved motifs of BjARF proteins.** (PNG)

**S4 Fig. Alignment of the Auxin_resp domain in BjARF proteins.** Multiple sequence alignment was performed using MAFFT version 7. (PNG)

## Author Contributions

**Conceptualization:** Wenbo Li, Jian Gao.

**Data curation:** Fabo Chen, Yinping Wang, Qinqin Yi.

**Formal analysis:** Haoyue Zheng.

**Funding acquisition:** Jian Gao.

**Investigation:** Fabo Chen.

**Methodology:** Wenbo Li, Yinping Wang, Yun Ren.

**Project administration:** Haoyue Zheng.

**Software:** Wenbo Li, Yun Ren.

**Validation:** Fabo Chen, Qinqin Yi.

**Writing – original draft:** Wenbo Li, Jian Gao.

**Writing – review & editing:** Wenbo Li, Fabo Chen, Yinping Wang, Haoyue Zheng, Qinqin Yi, Yun Ren, Jian Gao.

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
