## [Decision Letter · Decision Letter 0]

18 Feb 2020

PONE-D-19-34855

Genome-wide identification and functional analysis of ARF transcription factors in Brassica juncea var. tumida

PLOS ONE

Dear Dr. Gao,

Thank you for submitting your manuscript to PLOS ONE. After careful consideration, we feel that it has merit but does not fully meet PLOS ONE’s publication criteria as it currently stands. Therefore, we invite you to submit a revised version of the manuscript that addresses the points raised during the review process.

We would appreciate receiving your revised manuscript by Apr 03 2020 11:59PM. To enhance the reproducibility of your results, we recommend that if applicable you deposit your laboratory protocols in protocols.io, where a protocol can be assigned its own identifier (DOI) such that it can be cited independently in the future. For instructions see: http://journals.plos.org/plosone/s/submission-guidelines#loc-laboratory-protocols

We look forward to receiving your revised manuscript.

Kind regards,

Yutaka Oono

Academic Editor

PLOS ONE

Journal Requirements:

Reviewers' comments:

Reviewer's Responses to Questions

**Comments to the Author**

1. Is the manuscript technically sound, and do the data support the conclusions?

Reviewer #1: Yes

Reviewer #2: Yes

2. Has the statistical analysis been performed appropriately and rigorously? 

Reviewer #1: Yes

Reviewer #2: Yes

3. Have the authors made all data underlying the findings in their manuscript fully available?

Reviewer #1: Yes

Reviewer #2: Yes

4. Is the manuscript presented in an intelligible fashion and written in standard English?

Reviewer #1: Yes

Reviewer #2: Yes

5. Review Comments to the Author

Reviewer #1: This manuscript addressed to clarify the members of ARF genes in tumorous stem mustard, a popular vegetable in China, and their possible function based on the in silico analysis and experiments. The topic is interesting and the results obtained are useful for futher understanding the roles of ARF gene family in plants. The experiments were well designed, the data were presented in solid, and the manucript was well written. I think this manuscript is at a good shap and can be accepted to publish in PLoS ONE.

Reviewer #2: This manuscript describes detailed characterization of sequences and expression profiles of Auxin response factor (ARF) genes in Brassica juncea var tumida. The presentation of the data in this paper seems to follow authors’ previous paper “Genome-wide identification and characterization, phylogenetic comparison and expression profiles of SPL transcription factor family in B. juncea (Cruciferae)”. This manuscript uses the same original data and similar bioinformatics methods to the previous paper. However, targeted genes (SPLs or ARFs) for the analysis are different; Both SPLs and ARFs are transcriptional factor but there is no functional or structural relationship between them. I feel the present work has been written independently and present original results.

Although characterizing ARF family genes has been already done in many plant species, this work is still valuable for some scientists who are interested in evolution in Brassica species and also who need detailed information about ARF gene family.

I feel two major problems are in this manuscript;

First, the discussion part is largely filled by repeating the result section. The authors should explore their present data by comparing the data in their previous publication or other related articles. For example, authors may be able to discuss about difference and similarity between ARFs and SPLs in regard to the phylogenetic relationships in B. juncea, A. thaliana, and B. napus. Also it may be interesting to discuss how extent miR160/167c-mediated post-transcriptional regulation is conserved in plant species.

The second problem is that explanation for specialized words is insufficient. The manuscript is not very clear for a person who is not familiar with this plants, B. juncea as well as ARFs. For example, there is no explanation what is ARF. Also it is hard to understand what tumor stem is.

(other minor points)

L 52, a typo; “genes –.”

L151-153,　For reader without knowledge for this plant, it may be good to show illustrations or photos for seedling stage (SS), mature stage (MS), and flower stage (FS) with approximate developmental date. Or cite an appropriate reference if you just followed previous work.

What is YA1, 2, and 3? Are they just different individuals or name of cultivar? Please indicate clearly in the manuscript.

L165, For the IAA treatment, it is not clear how did you spray IAA to plants; keep to spray IAA for 8 hours, or sprayed IAA then keep 8 hours?

L188, explain here what is Auxin-resp domain.

Please be consistent with the terms used in the document, “a”uxin_resp domain or “A”uxin_resp domain.

L223-227,　In the text, it sounds that the subgroup is independent from group III. However, in figure 2 the subgroup is belong to Group III.

L 234, a typo; “in B .juncea” should be “in B. juncea”

L 270, colored boxs numbered 1 to 9 are NOT at the bottom.

L 295, no explanation for fig 5. What do the numbers as well as values of x-axis mean?

L 300, the authors should explain why they focused on these miRNA at the beginning of this section or in the introduction.

L 302, what is sub-group II? There is no sub-group II in figure 1. It maybe group II.

L 304, what is BjmiR160a-c? Is it same meaning to BjmiR160?

L310-303 Fig6 A and B. Labels seem to be wrong.

L328, explain why you used three different materials, YA1 (yonganxiaoye1), YA2 (yonganxiaoye2), and YA3.

L338, the authors should provide more information about developmental stage of this plants so that readers can image them (e.g. stem nodule of vegetal period (VP), flower period (FP) and etc).

L404. Although the authors mention that B.napa was more closely related to B juncea in evolution, it sounds totally obvious to require any analyses shown in this manuscript. More explanation of the implications of this findings should be provided.

L410, a typo, “showed that revealed that”

6. PLOS authors have the option to publish the peer review history of their article (what does this mean?). If published, this will include your full peer review and any attached files.

Reviewer #1: Yes: Kun-Ming Chen

Reviewer #2: No

---

## [Author Response · Author response to Decision Letter 0]

31 Mar 2020

Dear Editor,

We would like to resubmit the revised manuscript entitled “Genome-wide identification and functional analysis of ARF transcription factors in Brassica juncea var. tumida” for consideration of publication in “PLOS ONE”. We would like to thank the editor and reviewers for thoroughly reviewing our manuscript and making many thoughtful comments. We have revised the manuscript to address reviewers’ comments, described in detail below and noted them by highlighting in modified manuscript. The main corrections and responses in the paper are listed one by one as follows.

Best regards

Jian Gao on behalf of all authors

Email: Gaojian_genomics@163.com

Response to Reviewer 1: 

Reviewer #1: This manuscript addressed to clarify the members of ARF genes in tumorous stem mustard, a popular vegetable in China, and their possible function based on the in silico analysis and experiments. The topic is interesting and the results obtained are useful for futher understanding the roles of ARF gene family in plants. The experiments were well designed, the data were presented in solid, and the manucript was well written. I think this manuscript is at a good shap and can be accepted to publish in PLoS ONE.

Response: Thank for you thoroughly reviewing our manuscript, we were very pleased to see that you recognized the novelty and potential significance of our work.

Response to Reviewer 2: 

Q1: First, the discussion part is largely filled by repeating the result section. The authors should explore their present data by comparing the data in their previous publication or other related articles. For example, authors may be able to discuss about difference and similarity between ARFs and SPLs in regard to the phylogenetic relationships in B. juncea, A. thaliana, and B. napus. Also it may be interesting to discuss how extent miR160/167c-mediated post-transcriptional regulation is conserved in plant species.

Response: Thank for you thoughtful comments. We have expanded the discussion section about miR160/167-mediated post-transcriptional regulation in plants. As well as we deleted some repeat sentences of the result section.

Q2: The second problem is that explanation for specialized words is insufficient. The manuscript is not very clear for a person who is not familiar with this plants, B. juncea as well as ARFs. For example, there is no explanation what is ARF. Also it is hard to understand what tumor stem is.

Response: Thank for your suggestion and we apologize for our confusion. We have introduced the plant in introduction section (L92-L94). ARF means Auxin response factor (L47), an important transcription factors involved in both the auxin signaling pathway and the regulatory development of various plant organs. Tumor stem is the vegetative organ of Tumorous stem mustard (L94).

Q3: L 52, a typo; “genes –.”

Response: Thanks for your reminding, and we apologize for our mistake. We have retyped it as “genes” (L52). 

Q4: L151-153,　For reader without knowledge for this plant, it may be good to show illustrations or photos for seedling stage (SS), mature stage (MS), and flower stage (FS) with approximate developmental date. Or cite an appropriate reference if you just followed previous work.

What is YA1, 2, and 3? Are they just different individuals or name of cultivar? Please indicate clearly in the manuscript.

Response: Thanks for your reminding, and we apologize for our confusion. We have defined seedling stage (SS, 15-days after germination), mature stage (MS, 120-days after germination) and flower stage (FS, 180-days after germination) in the modified manuscript.

YA1, 2, and 3 are represent three different cultivars of Brassica juncea var. tumida which cultivated in China, and we have explained in the modified document (L155-157).

Q5: L165, For the IAA treatment, it is not clear how did you spray IAA to plants; keep to spray IAA for 8 hours, or sprayed IAA then keep 8 hours?

Response: Thank for your suggestion. We have provided a detailed method for IAA treatment. We sprayed the plant with 50μM IAA solution, and then keep it for eight hours until plant tissue collection (L171-174).

Q6: L188, explain here what is Auxin-resp domain.

Please be consistent with the terms used in the document, “a”uxin_resp domain or “A”uxin_resp domain.

Response: Thank for your suggestion. We have explained the Auxin-resp domain in modified manuscript (L195), it represents a conserved region of auxin-responsive transcription factors. And we have unified it as “Auxin_resp domain” in our manuscript (L247, L258).

Q7: L223-227,　In the text, it sounds that the subgroup is independent from group III. However, in figure 2 the subgroup is belong to Group III.

Response: Thank for your suggestion, and we apologize for our confusion. Our results showed that AtARF12-15, AtARF20-AtARF22 and ARFun1-3 were belong to Group III, but AtARF12-15, AtARF20-AtARF22 were categorized into a sub-group of Group III and ARFun1-3 were categorized into a novel sub-group of Group III (L233-234).

Q8: L 234, a typo; “in B .juncea” should be “in B. juncea”

Response: Response: Thanks for your reminding, and we apologize for our mistake. We have corrected it as “B. juncea” (L242). 

Q9: L 270, colored boxs numbered 1 to 9 are NOT at the bottom.

Response: Thanks for your reminding, and we apologize for our mistake. We have corrected it as “at the upper right corner” (L278-279).

Q10: L 295, no explanation for fig 5. What do the numbers as well as values of x-axis mean?

Response: Thank for your advice, and we apologize for our negligence. We have explained fig 5 in revised manuscript. Value of x-axis means the number of cis-acting elements which founded in BjARFs promoters (L304-306).

Q11: L 300, the authors should explain why they focused on these miRNA at the beginning of this section or in the introduction.

Response: Thank for you thoughtful comments. We have introduced these miRNA in introduction section (L88-91).

Q12: L 302, what is sub-group II? There is no sub-group II in figure 1. It maybe group II.

Response: Thanks for your reminding, and we apologize for our mistake. We have corrected it as “Group II” (L313).

Q13: L 304, what is BjmiR160a-c? Is it same meaning to BjmiR160?

Response: Thanks for your advice, BjmiR160a-c is same meaning to BjmiR160. We have rewritten BjmiR160a-c as BjmiR160 (L315).

Q14: L310-303 Fig6 A and B. Labels seem to be wrong.

Response: Thanks for your reminding, and we apologize for our mistake. We have corrected it in the text (L323-324).

Q15: L328, explain why you used three different materials, YA1 (yonganxiaoye1), YA2 (yonganxiaoye2), and YA3.

Response: Thank for you thoughtful comments. We selected YA1, YA2, and YA3 because they are three important inbred line of Brassica juncea var. tumida. A target gene may have different expression profiles in different cultivars, we used three materials to screen common candidate BjARF genes which may response to tumor stem development or other biology process.

Q16: L338, the authors should provide more information about developmental stage of this plants so that readers can image them (e.g. stem nodule of vegetal period (VP), flower period (FP) and etc).

Response: Thanks for your advice, we have described the developmental stage of Brassica juncea var. tumida in modified manuscript (L348-350).

Q17: L404. Although the authors mention that B.napa was more closely related to B juncea in evolution, it sounds totally obvious to require any analyses shown in this manuscript. More explanation of the implications of this findings should be provided.

Response: Thank for you thoughtful comments. In this study, we found that B.napa was more closely related to B juncea in evolution. And it suggested that gene function of unique genes are more similar between B. juncea and B. napa, we should give priority to B. napa as a reference for gene function research in future (L416-418).

Q18: L410, a typo, “showed that revealed that”

Response: Thanks for your reminding, and we apologize for our mistake. We have corrected it as “showed that” (L423).

---

## [Editor Report · Decision Letter 1]

7 Apr 2020

Genome-wide identification and functional analysis of ARF transcription factors in Brassica juncea var. tumida

PONE-D-19-34855R1

Dear Dr. Gao,

We are pleased to inform you that your manuscript has been judged scientifically suitable for publication and will be formally accepted for publication once it complies with all outstanding technical requirements. Please note that I put two very minor comments below. 

With kind regards,

Yutaka Oono

Academic Editor

PLOS ONE

Additional Editor Comments (optional):

L21-23：　The sentence does't seem grammatically correct. Do you mean "In this study, 65 B. juncea genes that encode ARF proteins were identified in the B. juncea whole-genome, classified into three phylogenetical groups and found....., "?

L305-306:　Please separate title and explanation. Only title should be written in bold.

---

## [Editor Report · Acceptance letter]

9 Apr 2020

PONE-D-19-34855R1 

Genome-wide identification and functional analysis of *ARF* transcription factors in *Brassica juncea* var. *tumida*

Dear Dr. Gao:

I am pleased to inform you that your manuscript has been deemed suitable for publication in PLOS ONE. Congratulations! Your manuscript is now with our production department. 

With kind regards,

on behalf of

Dr. Yutaka Oono 

Academic Editor

PLOS ONE